# Spread Through Air Spaces (STAS) Is an Independent Prognostic Factor in Resected Lung Squamous Cell Carcinoma

**DOI:** 10.3390/cancers14092281

**Published:** 2022-05-03

**Authors:** Sami Dagher, Abdulrazzaq Sulaiman, Sophie Bayle-Bleuez, Claire Tissot, Valérie Grangeon-Vincent, David Laville, Pierre Fournel, Olivier Tiffet, Fabien Forest

**Affiliations:** 1Department of Pathology, University Hospital of Saint Etienne, 42055 Saint Etienne, France; sami.dagher@etu.univ-st-etienne.fr (S.D.); david.laville@etu.univ-st-etienne.fr (D.L.); 2Department of Thoracic Surgery, University Hospital of Saint Etienne, 42055 Saint Etienne, France; abdulrazzaq.sulaiman@chu-st-etienne.fr (A.S.); olivier.tiffet@chu-st-etienne.fr (O.T.); 3Department of Pneumology, University Hospital of Saint Etienne, 42055 Saint Etienne, France; sophie.bayle@chu-st-etienne.fr; 4Lucien Neuwirth Cancer Institute, 42271 Saint Etienne, France; claire.tissot@ramsaysante.fr (C.T.); pierre.fournel@icloire.fr (P.F.); 5Department of Pneumology, Hospital of Roanne, 42300 Roanne, France; valerie.grangeon@ch-roanne.fr; 6Department of Molecular Biology of Solid Tumors, University Hospital of Saint Etienne, 42055 Saint Etienne, France

**Keywords:** squamous cell carcinoma, Spread Through Air Spaces, tumor grading, prognostic

## Abstract

**Simple Summary:**

There is no histoprognostic grading for lung squamous cell carcinoma (LUSC). The objective of this work was to identify prognostic factors from a retrospective cohort study of pulmonary squamous cell carcinomas. In this single-center retrospective study of 241 patients, all patients with LUSC who underwent surgical excision over a 12-year period were included. The presence of Spread Through Air Spaces (STAS) was correlated with tumor location (*p* < 0.001), pathological stage (*p* = 0.039), tumor differentiation (*p* = 0.029), percentage of necrosis (*p* = 0.004), presence of vascular and/or lymphatic emboli, budding (*p* = 0.02), single cell invasion (*p* = 0.002) and tumor nest size (*p* = 0.005). On multivariate analysis, only STAS > 3 alveoli (HR, 2.74; 95% CI, 1.18–6.33) was related to overall survival. In conclusion, extensive STAS is an independent factor of poor prognosis in LUSC. STAS is correlated with the presence of other poor prognostic factors such as emboli and pleural invasion and would reflect greater tumor aggressiveness.

**Abstract:**

Objective: There is no histoprognostic grading for lung squamous cell carcinoma (LUSC). Different prognostic factors have been described in the recent literature and are not always studied in parallel. Our objective was to search for morphological histopathological prognostic factors in LUSC. Materials and Methods: In this single-center retrospective study of 241 patients, all patients with LUSC who underwent surgical excision over a 12-year period were included. The primary endpoint was 5-year overall survival. Results: STAS was present in 86 (35.7%) patients. The presence of Spread Through Air Spaces (STAS) was correlated with tumor location (*p* < 0.001), pathological stage (*p* = 0.039), tumor differentiation (*p* = 0.029), percentage of necrosis (*p* = 0.004), presence of vascular and/or lymphatic emboli, budding (*p* = 0.02), single cell invasion (*p* = 0.002) and tumor nest size (*p* = 0.005). The percentage of tumor necrosis was correlated with the overall survival at 5 years: 44.6% of patients were alive when the percentage of necrosis was ≥50%, whereas 68.5% were alive at 5 years when the necrosis was <30% (*p* < 0.001). When vasculolymphatic emboli were present, the percentage of survival at 5 years was 42.5% compared to 65.5% when they were absent (*p* = 0.002). The presence of isolated cell invasion was correlated with a lower 5-year survival rate: 51.1% in the case of presence, versus 66% in the case of absence (*p* = 0.02). In univariate analysis, performance status, pathological stage pT or pN, pleural invasion, histopathological subtype, percentage of tumor necrosis, vasculolymphatic invasion, single-cell invasion, budding and tumor nest size correlated with the percentage of survival at 5 years. On multivariate analysis, only STAS > 3 alveoli (HR, 2.74; 95% CI, 1.18–6.33) was related to overall survival. Conclusion: In conclusion, extensive STAS is an independent factor of poor prognosis in LUSC. STAS is correlated with the presence of other poor prognostic factors such as emboli and pleural invasion and would reflect greater tumor aggressiveness.

## 1. Introduction

Lung squamous cell carcinomas (LUSC) are malignant tumors frequently encountered in thoracic pathology. These tumors have not benefited from the same therapeutic advances as lung adenocarcinomas (LUAD) because there is no targeted therapy for LUSC. Recently, different prognostic factors have been individualized in this type of tumor, but their respective weight is not known when integrating all these factors. It is therefore interesting to better specify the prognostic factors of pulmonary squamous cell carcinomas in order to better treat them.

The 2021 World Health Organization (WHO) classification of thoracic tumors does not incorporate histopathologic criteria to stratify the prognosis of patients with LUSC [1]. Only the TNM stage and the performance score seem interesting to stratify patient’s risk [1]. The 2021 classification still proposes a subdivision into keratinizing, non-keratinizing and basaloid subtypes of LUSC, although it does not correlate with prognosis [1].

There has been an increased interest in histopathological prognostic factors in lung cancer in recent years. For LUAD, a grading system has been proposed and recommended by the WHO classification [1,2]. In contrast, for LUSC, various prognostic factors have been suggested but are still not validated in clinical practice. Of all the prognostic factors described, Spread Through Air Spaces (STAS) is a prognostic factor in LUAD, neuroendocrine carcinoma and metastases to the lung of colon cancer [3,4,5]. In the literature, few series have studied STAS in lung squamous cell carcinoma [6,7,8,9,10,11].

Nevertheless, other prognostic factors have been described in the literature such as percentage of surface area occupied by necrosis with a worse prognosis when the percentage of necrosis is high [12,13,14]. A high mitotic count has, paradoxically, been shown to be a good prognostic factor in LUSC [12,15,16]. In two works with common authorship, the prognostic value of nuclear size has been shown to be a factor decreasing recurrence-free survival [15,17]. The tumor-stroma ratio is known to be a prognostic factor in other tumors such as esophageal squamous cell carcinoma [18]. In lung cancers, a study of 261 NSCLC, including 79 LUSC, showed a decreased overall survival in patients with stroma-rich tumors [19]. Isolated tumor cell infiltration and budding at the periphery of the tumor have been shown to be a poor prognostic factor [15,17]. Tumor nest size has been shown to have prognostic value in squamous cell carcinomas of other origins such as cervical carcinoma [20]. Classical histopathological criteria, such as differentiation, basaloid architecture and keratinization are included in the WHO classification as subtypes despite the absence of prognostic value [1]. A grading system has been suggested but not internationally accepted [17]. Nevertheless, some of the proposed prognostic factors have not been found by other studies and are not universally recognized.

We therefore investigated in a cohort of operated LUSC which histopathological factors could have a prognostic impact on 5-year survival.

## 2. Materials and Methods

### 2.1. Patients

This work has been approved by the Ethic Committee of the University Hospital Center of Saint Etienne (Terres d’Ethique, IRBN1112020/CHUSTE obtained in July 2020). This is a retrospective cohort study including all patients with a diagnosis of stage I to III LUSC operated on at our center between 2008 and 2020. All tumors were reclassified according to the WHO 2021 classification of thoracic tumors. Patients who received neoadjuvant chemotherapy were not included.

The following clinical data were collected: date of birth, date of diagnosis, smoking status, type of surgical specimen, number of metastatic lymph node sites, type of adjuvant therapy received. Overall survival was calculated from the date of diagnosis to the date of death or last news. All tumors were restaged according to the AJCC 8th Edition [21].

### 2.2. Histopathologic Evaluation

Tumor slides were examined separately and then jointly in case of disagreement by two pathologists blinded to clinical outcome (SD, FF). The mean number of slides examined per case was 6.2 ± 0.1. The subtypes collected were the keratinizing, nonkeratinizing and basaloid subtypes as specified in the WHO classification [1].

The relationship between the pleura and the tumor was evaluated as previously defined: no extension beyond the visceral elastic boundary for PL0, invasion beyond the elastic layer for PL1, invasion of the pleural surface for PL2 and invasion of the parietal pleura for PL3 [22].

The presence of lymphatic or vascular emboli was defined by the presence of clusters of tumor cells in the lymphatic or venous vessels, respectively. Mitotic count was assessed on the area with the largest number of mitoses selected at ×100 magnification and then counted on 2 mm^2^ at ×400.

The tumor-stroma ratio was defined as the area occupied by the stroma compared to the total tumor area.

The localization of the tumor was recorded as central, intermediate and peripheral tumors: peripheral tumors are defined as tumors arising in the outer third of the lung. Central tumors are defined as tumors arising from lobar bronchus. The presence of carcinoma in situ was recorded on the adjacent bronchus. Carcinoma in situ was not considered for the measurement of tumor diameter. The presence of obstructive pneumonitis related to the presence of the tumor has been recognized. We recorded the presence of peritumoral pneumopathy defined by the presence of pneumopathy lesions in contact with the tumor without an obstructive lesion in a bronchus.

Peripheral infiltration was assessed by different methods previously described [23,24]. In 10 consecutive areas, at ×200 magnification, the number of clusters of 5 cells or less were counted in the highest areas. These clusters, equal or less than 5 cells, were defined as “buds”. Buds were scored as previously described: BD0 for 0 bud/10HPF, BD1 for 1 to 4 buds in 10/HPF, BD2 for 5 to 9 buds and BD3 when ≥10 buds where found.

The size of the smallest tumor nest present at the periphery of the tumor was also quantified. This size is defined as the size of the smallest invasive tumor cell nest at the tumor periphery. Tumor nest size was assessed by scanning the slides at ×100 on all tumor slides. This tumor size was categorized as large when the smallest tumor nest was equal to at least 15 cells, intermediate when the smallest nest was greater than or equal to 5 cells and less than 15 cells, small for 2 to 4 cells and isolated tumor cells when the smallest nest consisted of only one cell.

The nuclear diameter was evaluated as previously described [17]. Briefly, on 3 fields at ×400 magnification, nuclear diameter was assessed as large when the nucleus was larger than the size of 4 resting lymphocytes, or small when the nuclear size was less than or equal to that of 4 resting lymphocytes.

STAS was defined by the presence of free-floating cell clusters within the alveoli at the periphery of the tumor. Due to a lack of consensus, STAS was assessed by two methods. The first method was to measure the distance in mm between the farthest intra-alveolar cell cluster and the tumor. The second method was to count the number of free alveolar spaces between this cluster and the tumor. We have grouped STAS as previously described in limited or extensive STAS [6]. Illustrative microphotographs are shown in Figure 1.

### 2.3. Statistical Analysis

Statistical analysis was performed with R software for windows and Rstudio for windows [25]. The package survival was used for survival analysis [26]. Descriptive statistics such as mean and standard deviation for continuous variables are provided. Exact Test of Fisher or χ^2^ was used for the categorical variable. Pearson’s test was used for quantitative variables. Results are reported as two-sided Ps and/or 95% confidence intervals. Failed or missing data are not included in statistical analysis. Cohen’s weighted kappa was calculated to assess the reproducibility in the two observers between categorical variables. The Shapiro-Wilk test was used to test normality for continuous variables. To assess the relationship between continuous variables when the normality was not present, Spearman test was used and ρ are given; the Pearson correlation test was used when normality was present and R2 were given.

Overall survival was calculated by the Kaplan–Meier method, and log-rank *p* values were used to assess significance. Survival curves were constructed with the Kaplan–Meier method. The date of diagnosis to the date of death (or censoring if the patient was alive at the time of last follow-up) was used to calculate overall survival. Median overall survival and upper and lower confidence intervals (CIs) are given. All reported *p* values are two-tailed; *p* < 0.05 was considered to be significant. Prognostic significance of clinical and pathological characteristics with *p* < 0.25 with log rank test were analyzed with univariate Cox regression and multivariate Cox model to determine hazard ratios (HRs).

## 3. Results

### 3.1. Patient Characteristics

Two hundred forty-one patients with a mean age at diagnosis of 68 ± 0.6 years were included. More than 90% of patients were males, 239 were smokers or formerly smokers, for two patients the history of smoking was not specified. Seventy-two patients underwent pneumonectomy, 19 underwent bilobectomy, 1 patient underwent lobectomy and segmentectomy, 127 underwent lobectomy, 3 patients underwent lobectomy and chest wall resection, 3 patients underwent segmentectomy and 16 underwent wedge resection. Median follow-up was 24.2 months (range: 0.01–143 months). Ninety-four patients (39%) died within 5 years after surgery. Sixty-two patients were treated with adjuvant chemotherapy and/or radiotherapy. The clinical and pathological characteristics are summarized in Table 1.

### 3.2. Histopathological Features

STAS was present in 56 cases out of 113 peripheral or intermediate tumors, it was present in 30 cases out of 128 central tumors, there was a correlation between the presence of STAS and the location of the tumor (*p* < 0.001). The presence of STAS was correlated with the pathological stage (*p* = 0.013), it was present in 42 out of 93 patients for stage I, in 16 out of 70 patients for stage II and 25 out of 70 patients for stage III. The presence of STAS correlated with the presence of vasculolymphatic emboli. STAS was present in 61 of 194 patients without emboli, in 25 of 47 patients with emboli (*p* = 0.005). The presence of STAS was correlated with the presence of budding. STAS was present in 39 of 79 BD0, in 27 of 78 BD1, in 14 of 51 BD2 and in 6 of 34 BD3 (*p* = 0.004). The presence of STAS was correlated with the presence of single cell invasion. STAS was present in 67 of 153 patients without single-cell invasion, in 19 of 88 patients with single-cell invasion (*p* < 0.001). The presence of STAS correlated with the size of the smallest cell nest, STAS was present in 19 out of 88 patients with a nest size of one cell, in 30 out of 86 patients with a nest size >1–≤5, and in 27 out of 55 patients with a nest size > 5 (*p* = 0.001).

The details of extended or limited STAS is presented in Table 2.

### 3.3. Correlation of Overall Survival with Clinicopathologic Data

The correlation between clinical data and survival is shown in Table 1 and Figure 2. Twenty-one patients were lost to follow-up. Clinical and pathological factors influencing 5-year overall survival were pT, pN, pathological stage, pleural invasion, histopathologic type, percentage of tumor necrosis, vascular emboli, cell nest group size and the infiltration pattern categorized either by the tumor budding classification by the presence of single cell invasion (Table 1). The 5-year overall survival rate was 76.9% for pT1, 59% for pT2, 54.7% for pT3 and 36.7% for pT4 (*p* < 0.001). Patients with pN2 had a significantly lower 5-year overall survival rate at 33.4% than pN0 and pN1 patients with 5-year overall survival rate at 66.4% and 63.3%, respectively (*p* = 0.006). Pathological stage calculated from pT and pN was significantly correlated with overall survival (*p* = 0.002). The 5-year overall survival rate was 75.3%, 58.6% and 47.1% for stage I, II and III, respectively. Pleural invasion was correlated with 5-year overall survival (*p* = 0.02). Patients with PL3 or PL1/PL2 pleural invasion had a 5-year overall survival rate of 50% and 48.1%, respectively. Patients without pleural invasion had a 5-year overall survival rate at 65.5%. Histopathologic subtype correlated with 5-year survival rate (*p* = 0.04). Keratinizing and non-keratinizing subtypes had a 5-year survival rate of 59.1% and 59.3%, whereas patients with a basaloid subtype had a survival rate of 92.3%. The percentage of surface area occupied by necrosis correlated with 5-year survival rate (*p* < 0.001). The 5-year survival rate was 44.6%, 53.8% and 68.5% for patients with ≥50%, ≥30%–< 50% and <30% tumor necrosis, respectively.

Sex, age, tumor side, tumor location, number of metastatic node sites, peritumoral or obstructive lung disease, quality of resection, tumor differentiation, tumor-stroma ratio, nuclear diameter, presence of carcinoma in situ, and presence of STAS measured with two methods and mitotic count were not correlated with 5-year survival.

The performance status showed a trend for a correlation to 5-year survival rate at 25% for PS3-4, 65% for PS1-2 and 68.3% for PS0.

### 3.4. Association between Clinicopathologic Factors and Survival

To identify independent prognostic factors associated with 5-year overall survival, univariate and multivariate Cox proportional hazard regression models were used. In univariate and multivariate analysis, only parameters with a *p* < 0.25 with the log-rank test were retained for the univariate test. The multivariate analysis retained all parameters with a *p* < 0.25 with the log-rank test in order to retain all parameters with a potential role, even below the significance thresholds, in order to best correlate the different parameters including STAS to prognosis. In univariate analysis, performance status, pT stage, pN stage, pathological stage, pleural invasion, histopathological subtype, the percentage of tumor necrosis, vascular and/or lymphatic emboli, single cell invasion, tumor budding and cell nest size were related to overall survival (Table 3).

The hazard ratios (HR) were calculated in a multivariable Cox model. On multivariate analysis, only STAS >3 alveoli (HR, 2.74; 95% CI, 1.18–6.33) was related to overall survival (Table 4).

### 3.5. Subgroup Analysis in Stage I and II Patients Who Underwent Lobectomy

We performed a subgroup analysis of patients who were treated with lobectomy and stage I–II. The main results are presented in Table 5. Prognostic factors related to 5-year survival rate were pT stage, pathological stage and percentage of tumor necrosis (*p* < 0.05). Budding and the presence of cell nests is at the limit of significance.

## 4. Discussion

Our study of 241 LUSC focused on the analysis of different morphological factors identified in the literature. The analysis of tumor morphology for prognosis is used in particular in current practice for breast carcinomas and prostate adenocarcinomas. The fifth edition of the WHO classification published in 2021 does not recommend a grading for LUSC, the main prognostic factors are performance status and tumor stage.

For several years, the prognostic value of STAS has been the subject of several works in lung adenocarcinoma. In these different studies, the definition of STAS extension was not consensual and was defined either by a measurement of the number of alveoli from the tumor edge or by a measurement in mm.

In LUSC, we found four series evaluating STAS. Kadota et al. showed in a series of 216 LUSC that STAS conferred an increased risk of locoregional and distant recurrence [7]. STAS was associated with more lymph node metastasis, higher pathological grade, more lymphatic emboli and more high-grade budding. There was no prognostic value on 5-year overall survival. In a series of 445 LUSC, Lu et al. showed that STAS was associated with more locoregional recurrence, more distant recurrence and greater mortality in stage II and III [6]. Yanagawa et al. showed in a cohort of 220 LUSC that STAS was a poor prognostic factor decreasing recurrence-free survival independently of stage and decreasing overall survival in stage I LUSC but not in stages II and III [8]. Neppl et al. found no prognostic value of STAS on overall survival and recurrence-free survival in their cohort of 354 LUSC [24]. In these four studies, the authors used the same definition of STAS and reported a frequency of STAS between 19% and 40% of cases. The authors found no significant difference in overall survival between limited STAS and STAS beyond the third alveolar space. We studied extensive and limited STAS, originally described in lung adenocarcinoma [3]. STAS is sometimes criticized and the question of its artifactual origin arises, and several authors have tried to prove that it is a phenomenon present in vivo [27]. It has been shown that STAS can be connected to the tumor [28,29]. We attempted to limit this bias by distinguishing between extensive and limited STAS.

In our work, we show that the presence of STAS is an independent factor decreasing the 5-year overall survival rate. Our work shows no prognostic value of STAS when measured in mm whereas it has a prognostic value when measured in number of alveolar spaces. This may be due to the variability of fixation protocols between different laboratories inducing a variable expansion of alveoli. Nevertheless, the alveolar count, although having a prognostic value, may seem inaccurate in patients who are often smokers, with emphysema and therefore a variability of alveolar size. Our statistical analysis shows no correlation in univariate analysis, but shows a correlation between survival and the presence of STAS in multivariate analysis. As multivariate analysis allows to take into account the weight of different parameters, it allows to correct the weight given to some histological parameters. Moreover, our approach allows to take into account parameters for which univariate analysis could lack statistical power because of a too small number of patients.

We observed a correlation between STAS and several parameters of aggressiveness such as the presence of lymphatic emboli, tumor stage and pleural invasion. These findings provide an additional argument against the artifactual origin of STAS and reinforce the idea of a higher tumor aggressiveness in the presence of STAS. Recently, it has been shown that the presence of STAS is associated with a decrease in E-cadherin expression in adenocarcinomas [11]. E-cadherin is involved with other proteins in the epithelial-mesenchymal transition process as shown in tumor budding for LUSC [30]. Moreover, tumor emboli, a phenomenon probably related to the epithelial-mesenchymal transition process, is correlated with the presence of STAS in our work [31]. STAS, tumor budding and tumor emboli could represent morphological translations of the same epithelial-mesenchymal transition phenomenon.

In 2016, the International Tumor Budding Consensus Conference proposed a consensus for the evaluation of budding in colorectal adenocarcinoma [23]. This grading was validated on LUSC showing that it was a poor prognostic factor on 354 LUSC showing that budding is an independent prognostic marker [24]. Kadota et al. analyzed a cohort of 458 LUSC and showed that infiltration by independent cells and nuclear diameter were independently correlated with survival of patients with LUSC [17]. Nevertheless, concerning budding, the authors had found a borderline significance suggesting that external validation might be needed for budding as a prognostic factor of LUSC [17]. Our work showed that budding is a prognostic factor in univariate analysis, but not in multivariate analysis. Nevertheless, our correlation analysis shows that budding was associated with other aggressiveness factors such as pT, pN, stage, tumor necrosis, vascular emboli and STAS. For the evaluation of budding, we did not use immunohistochemistry even in cases where its evaluation was difficult. Nevertheless, Neppl et al. evaluated the value of immunohistochemistry with an anti-cytokeratin antibody and found no difference in classification of budding with immunohistochemistry with good reproducibility [24].

Weichert et al. analyzed budding on 541 LUSCs and showed that the size of the smallest focus of infiltrating cells correlated with overall survival [32]. However, cell-independent infiltration was not correlated with overall survival [32]. The definition of independent cells, or the size of the smallest infiltrating focus, partially overlaps with the definition of budding [23,32]. In our work, the presence of independent cells was significantly correlated with overall survival in univariate analysis but not in multivariate analysis.

Subtyping of LUSCs into keratinizing and non-keratinizing did not show prognostic value in our work as demonstrated by others [15]. The assessment of the prognostic value of the basaloid contingent has little value in our work because of the small number of patients. The prognostic value of the basaloid contingent is unclear in the literature [15,33].

In a study of 261 NSCLCs including 79 LUSCs, it was shown that a tumor-stroma ratio greater than 50% induced decreased overall survival [19]. Nevertheless, in our work on a homogeneous cohort of the same histological type we did not find any prognostic value of this factor in multivariate analysis.

Several studies have shown that high mitotic activity is a good prognostic factor [12,15,16]. However, we did not find this association in our work.

Two studies of 178 NSCLC including 111 LUSC and 76 LUSC showed that the presence of necrosis was an independent factor of poor prognosis [13,14]. We find a prognostic value in univariate analysis but not in multivariate analysis of the extent of necrosis.

In our work, the 5-year overall survival is comparable to other studies: 70% for stage I, 57% in stage II and 21% in stage III/IV, in the work of Kadota et al. [17].

Among the limitations of this work, its retrospective nature is an inherent limitation with some missing data. However, despite the long study period of our work, it provides sufficient statistical power to identify prognostic factors. We minimized bias with a double reading and consensus meeting, as well as with a broad sampling of our tumors. In addition, we examined all tumor slides and not just one representative slide. Finally, our study was interested in incorporating most of the recently described prognostic factors in LUSC. Another limitation of our work is that our primary endpoint is 5-year survival; we could have assessed progression-free survival. Nevertheless, overall survival is a more important criterion than progression-free survival in the evaluation of prognosis in lung cancer. A limitation is that the prognostic value is not found in the group of patients treated by lobectomy and stage I–II. This subgroup analysis is limited due to the number of patients and does not allow to identify these factors as significant.

In conclusion, we showed that extensive STAS was an independent factor of poor prognosis. Other factors have prognostic value in univariate analysis, but not in multivariate analysis. On the other hand, STAS is correlated with the presence of other poor prognostic factors such as emboli and pleural invasion and would reflect greater tumor aggressiveness. STAS is not yet included in the therapeutic decision of LUSC, but could be included in pathology reports to provide additional information to clinicians.

## Figures and Tables

**Figure 1 cancers-14-02281-f001:**
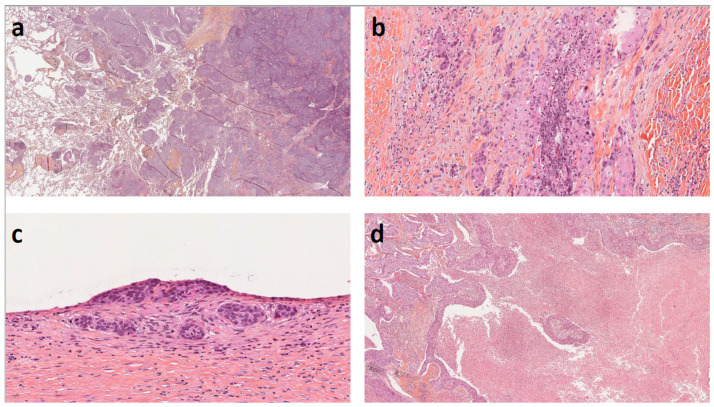
Illustrative microphotographs of histopathological parameters: (**a**)—Hematoxylin, Eosin and Saffron (HES) × 25, STAS. (**b**)—HES × 100, Tumor budding and single cell invasion. (**c**)—HES × 200, Pleural invasion PL2. (**d**)—HES × 25, Extensive tumor necrosis.

**Figure 2 cancers-14-02281-f002:**
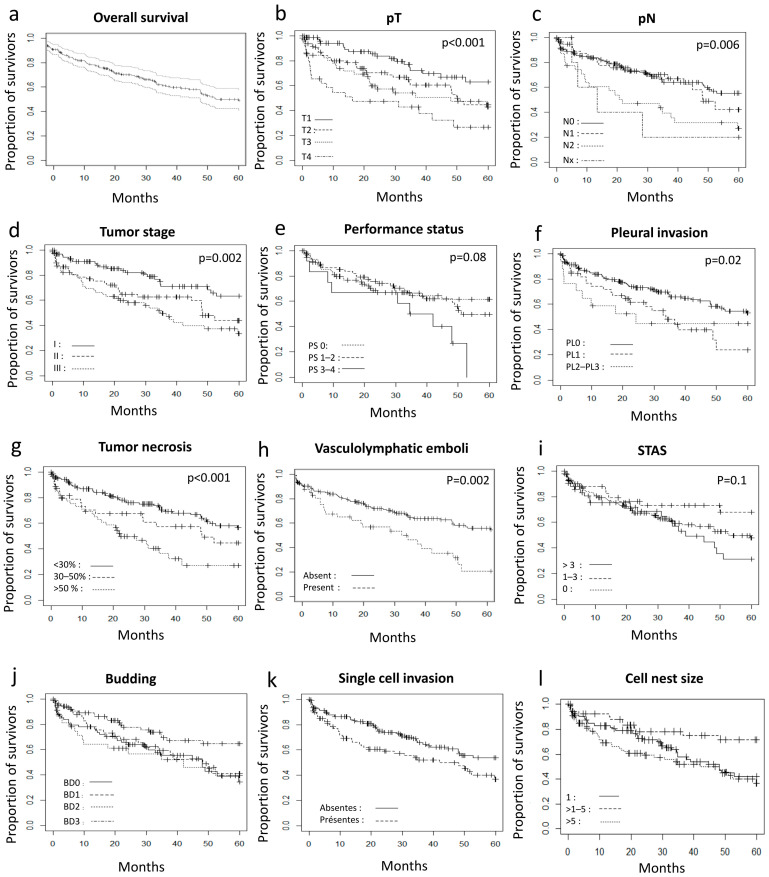
Kaplan–Meier curves for 5-year overall survival according to clinical and pathological parameters. (**a**)—Overall Survival with 95% confidence intervals. (**b**)—pT stages. (**c**)—pN stages. (**d**)—Tumor stage. (**e**)—Performance status. (**f**)—Pleural invasion. (**g**)—Tumor necrosis. (**h**)—Vasculolymphatic emboli. (**i**)—STAS. (**j**)—Tumor budding. (**k**)—Single cell invasion. (**l**)—Cell nest size.

**Table 1 cancers-14-02281-t001:** Clinical and histopathological data in relation to 5-year overall survival.

Variable	n (%)	5-Year Survival Rate	*p* (Log-Rank Test)
**Sex**			0.9
M	220 (91.3)	60.4%	
F	21 (8.7)	61.9%	
**Age**			0.9
≤65 years	99 (41.1)	58.6%	
>65 years	142 (58.9)	62%	
**PS**			0.08
0	79 (32.8)	68.3%	
1–2	80 (33.2)	65%	
3–4	12 (5)	25%	
Unknown	70 (29)	52.9%	
**Tumor side**			0.5
Right	127 (52.7)	59.8%	
Left	114 (47.3)	61.4%	
**Location**			0.1
Peripheral or intermediate	113 (46.9)	58.4%	
Central	128 (47.3)	62.5%	
**pT**			**<0.001**
pT1	78 (32.4)	76.9%	
pT2	78 (32.4)	59%	
pT3	52 (21.6)	54.7%	
pT4	30 (12.4)	36.7%	
Not assessable	2 (0.8)	50%	
**pN**			**0.006**
pN0	143 (59.3)	66.4%	
pN1	60 (24.9)	63.3%	
pN2	32 (13.3)	33.4%	
pNx	6 (2.5)	33.3%	
**Pathological Stage**			**0.002**
I	93 (38.6)	75.3%	
II	70 (29)	58.6%	
III	70 (29)	47.1%	
Not assessable	8 (3.3)	37.5%	
**Number of metastatic lymph nodes station**			0.1
0	143 (59.3)	64.4%	
1	64 (26.5)	56.2%	
≥2	28 (11.6)	46.4%	
Unknown	6 (2.5)	33%	
**Pleural invasion**			**0.02**
PL0	177 (73.4)	65.5%	
PL1 or PL2	52 (21.6)	48.1%	
PL 3	12 (5)	50%	
**Peritumoral pneumonitis**			0.3
Absent	150 (62.2)	62.7%	
Present	91 (37.8)	57.1%	
**Obstructive pneumonitis**			0.8
Absent	235 (97.5)	60.4%	
Present	6 (2.5)	66.7%	
**Type**			**0.04**
Keratinizing	93 (39.6)	59.1%	
Not keratinizing	135 (56)	59.3%	
Basaloid	13 (5.4)	92.3%	
**Quality of the resection**			0.5
R0	228 (94.6)	61.4%	
R1	13 (5.4)	46.1%	
**Differentiation**			0.8
Poorly	65 (27)	63.1%	
Moderately	125 (51.9)	60%	
Well	51 (21.2)	58.8%	
**Tumor to stroma ratio**			0.7
<30%	94 (39)	59.6%	
≥30% et <50%	72 (29.9)	59.9%	
≥50%	75 (31.1)	65.3%	
**Percentage of tumor necrosis**			**<0.001**
<30%	146 (60.6)	68.5%	
≥30% et <50%	39 (16.2)	53.8%	
≥50%	56 (23.3)	44.6%	
**In situ carcinoma**			0.6
Present	47 (19.5)	63.8%	
Absent	194 (80.5)	59.8%	
**Vascular and/or lymphatic emboli**			**0.002**
Absent	194 (80.5)	65.5%	
Present	47 (19.5)	42.5%	
**STAS**			0.1
Absent	154 (63.9)	59.7%	
≤3 alveoli	44 (18.2)	75%	
>3 alveoli	42 (17.4)	50%	
Not assessable	1 (0.4)	0%	
**STAS**			1
Absent	154 (63.9)	59.7%	
≤1 mm	55 (22.8)	61.8%	
>1 mm	32 (13.3)	62.5%	
**Budding**			**0.04**
Zero	58 (24)	73.4%	
BD1	40 (16.6)	53.3%	
BD2	29 (12)	56.9%	
BD3	17 (7)	50%	
Not assessable	2 (0.8)	100%	
**Single cell invasion**			**0.02**
Absent	153 (63.5)	66%	
Present	88 (36.5)	51.1%	
**Cell nests group**			**0.006**
1	88 (36.5)	52.3%	
≥1–<5	97 (40.2)	58.8%	
≥5	55 (22.8)	78.2%	
Not assessable	1 (0.4)	100%	
**Nuclear Diameter**			0.7
Large	112 (46.5)	61.6%	
Small	129 (53.5)	59.7%	
**Mitoses**			0.3
<20/2 mm^2^	211 (87.5)	59.2%	
≥20/2 mm^2^	29 (12)	69%	
Not assessable	1 (0.4)	100%	

M: Male, F: Female. PS: Performance Status. pT (Primary tumor characteristics). pN (Lymph nodes characteristics). PL (Pleural invasion). STAS (Spread Through Air Spaces). BD: Tumor Budding group. Number in bold indicate statistically significant results.

**Table 2 cancers-14-02281-t002:** Relationship between STAS and clinical and pathological parameters.

STAS	Absent	≤3 Alveoli	>3 Alveoli	Not Assessable	*p*
**Sex**					0.2 *
M	143 (59.3)	37 (15.3)	39 (16.2)	1 (0.4)	
F	11 (4.6)	7 (2.9)	3 (1.2)	0 (0)	
**Age**					0.472
≤65 years	68 (28.2)	16 (6.6)	15 (6.2)	0 (0)	
>65 years	86 (35.7)	28 (11.6)	27 (11.2)	1 (0.4)	
**PS**					0.293 *
0	50 (20.7)	18 (7.5)	11 (4.6)	0 (0)	
1–2	43 (17.8)	18 (7.5)	19 (7.9)	0 (0)	
3–4	10 (4.1)	1 (0.4)	1 (0.4)	0 (0)	
Unknown	51 (21.2)	7 (2.9)	11 (4.6)	1 (0.4)	
**Localization**					**<0.001**
Peripheral or intermediate	57 (23.6)	31 (12.9)	25 (10.4)	0 (0)	
Central	97 (40.2)	13 (5.4)	17 (7.1)	1 (0.4)	
**pT**					0.414 *
pT1	44 (18.3)	19 (7.9)	14 (5.8)	1 (0.4)	
pT2	50 (20.7)	13 (5.4)	15 (6.2)	0 (0)	
pT3	40 (16.6)	7 (2.9)	6 (2.5)	0 (0)	
pT4	18 (7.5)	5 (2.1)	7 (2.9)	0 (0)	
Not assessable	2 (0)	0 (0)	0 (0)	0 (0)	
**pN**					0.762
pN0	90 (37.3)	24 (10)	28 (11.6)	1 (0.4)	
pN1	42 (17.4)	10 (4.1)	8 (3.3)	0 (0)	
pN2	19 (7.9)	7 (2.9)	6 (2.5)	0 (0)	
pNx	0 (0)	0 (0)	0 (0)	0 (0)	
**Pathological Stage**					**0.039**
I	50 (20.7)	20 (8.3)	22 (9.1)	1 (0.4)	
II	54 (22.4)	10 (4.1)	6 (2.5)	0 (0)	
III	45 (18.7)	11 (4.6)	14 (5.8)	0 (0)	
Not assessable	5 (2.1)	3 (1.2)	0 (0)	0 (0)	
**Number of metastatic lymph nodes station**					0.662
0	89 (36.9)	24 (10)	28 (11.6)	1 (0.4)	
1	45 (18.7)	11 (4.6)	8 (3.3)	0 (0)	
≥2	16 (6.6)	6 (2.5)	6 (2.5)	0 (0)	
Unknown	4 (1.7)	3 (1.2)	0 (0)	0 (0)	
**Pleural invasion**					0.402 *
PL0	111 (46.1)	34 (14.1)	31 (12.9)	1 (0.4)	
PL1 or PL2	32 (13.3)	10 (4.1)	10 (4.1)	0 (0)	
PL3	11 (4.6)	0 (0)	1 (0.4)	0 (0)	
**Type**					0.079
Keratinizing	68 (28.2)	15 (6.2)	10 (4.1)	0 (0)	
Non-keratinizing	78 (32.4)	28 (11.6)	28 (11.6)	1 (0.4)	
Basaloïd	8 (3.3)	1 (0.4)	4 (1.7)	0 (0)	
**Quality of the resection**					0.374 *
R0	143 (59.3)	43 (17.8)	41 (17)	1 (0.4)	
R1	11 (4.6)	1 (0.4)	1 (0.4)	0 (0)	
**Differenciation**					**0.029** *
Poorly	41 (17)	7 (2.9)	17 (7.1)	0 (0)	
Moderate	74 (30.7)	30 (12.4)	20 (8.3)	1 (0.4)	
Well	39 (16.2)	7 (2.9)	5 (2.1)	0 (0)	
**Tumor to stroma ratio**					0.845
<30%	56 (23.2)	20 (8.3)	17 (7.1)	1 (0.4)	
≥30% et <50%	49 (20.3)	11 (4.6)	12 (5)	0 (0)	
≥50%	49 (20.3)	13 (5.4)	13 (5.4)	0 (0)	
**Percentage of tumor necrosis**					**0.004** *
<30%	86 (35.7)	24 (10)	35 (14.5)	1 (0.4)	
≥30% et <50%	27 (11.2)	11 (4.6)	1 (0.4)	0 (0)	
≥50%	41 (17)	9 (3.7)	6 (2.5)	0 (0)	
**Vascular and/or lymphatic emboli**					**0.02** *
Absent	132 (54.8)	32 (13.3)	29 (12)	1 (0.4)	
Present	22 (9.1)	12 (5)	13 (5.4)	0 (0)	
**BD**					**0.02** *
Zero	39 (16.2)	21 (8.7)	18 (7.5)	1 (0.4)	
BD1	48 (19.9)	14 (5.8)	13 (3.7)	0 (0)	
BD2	37 (15.3)	5 (2.1)	9 (3.7)	0 (0)	
BD3	28 (11.6)	4 (1.7)	2 (0.8)	0 (0)	
Not assessable	2 (0.8)	0 (0)	0 (0)	0 (0)	
**Single cell invasion**					**0.002**
Absent	85 (35.3)	34 (14.1)	33 (13.7)	1 (0.4)	
Present	69 (28.6)	10 (4.1)	9 (3.7)	0 (0)	
**Cell nests group**					**0.005**
1	69 (28.6)	10 (4.1)	9 (3.7)	0 (0)	
>1–≤5	56 (23.2)	18 (7.5)	22 (9.1)	1 (0.4)	
>5	28 (11.6)	16 (6.6)	11 (4.6)	0 (0)	
Not assessable	1 (0.4)	0 (0)	0 (0)	0 (0)	
**Nuclear diameter**					0.832
Large	74 (30.7)	19 (7.9)	19 (7.9)	0 (0)	
Small	80 (33.2)	25 (10.4)	23 (9.5)	1 (0.4)	
**Mitotic activity**					0.845
<20/2 mm^2^	134 (55.6)	38 (15.8)	38 (15.8)	1 (0.4)	
≥20/2 mm^2^	19 (7.9)	6 (2.5)	4 (1.7)	0 (0)	
Not assessable	1 (0.4)	0 (0)	0 (0)	0 (0)	

M: Male, F: Female. PS: Performance Status. pT (Primary tumor characteristics). pN (Lymph nodes characteristics). PL (Pleural invasion). STAS (Spread Through Air Spaces). BD: Tumor Budding group. *: Fisher’s exact test. Number in bold indicate statistically significant results.

**Table 3 cancers-14-02281-t003:** Univariate analysis of survival for clinical and pathological parameters.

Variable	IC95%	HR	*p* (Pr > z)
**PS**			
1–2	0.64–1.88	1.09	0.745
3–4	1.06–4.92	2.29	**0.034**
**Location (peripheral or intermediate)**	0.92–2.07	1.38	0.119
**pT**			
pT2	1.02–3.25	1.82	**0.042**
pT3	1.18–4	2.17	**0.013**
pT4	1.90–6.91	3.62	**<0.001**
**pN**			
pN1	0.68–1.89	1.13	0.630
pN2	1.32–3.69	2.21	**0.002**
**Pathological Stage**			
II	1.09–3.27	1.89	**0.022**
III	1.485– 4.21	2.50	**>0.001**
**Number of metastatic lymph nodes station**			
1	0.87–2.25	1.40	0.162
≥2	0.96–3.01	1.71	0.069
**Pleural invasion**			
PL1–PL2	1.19–2.96	1.88	**0.007**
PL3	0.74–3.97	1.72	0.210
**Histopathological subtype**			
Not keratinizing	0.64–1.47	0.97	0.905
Basaloid	0.02–0.87	0.12	**0.036**
**Percentage of tumor necrosis**			
≥30% et <50%	0.89–2.67	1.54	0.119
≥50%	1.69–4.28	2.69	**<0.001**
**Vascular and/or lymphatic emboli present**	1.29–3.16	2.01	**0.002**
**Single cell invasion present**	1.095–2.47	1.64	**0.017**
**STAS**			
≤3 alveoli	0.32–1.15	0.60	0.123
>3 alveoli	0.79–2.14	1.30	0.297
**BD**			
BD1	1.18–3.49	2.03	**0.010**
BD2	0.97-.3.22	1.77	0.062
BD3	1.13–4.15	2.16	**0.020**
**Cell nests group ≥ 5**			
≥1–<5	0.67–2.51	1.30	0.431
1	1.23–3.65	2.12	**0.006**

PS: Performance Status. pT (Primary tumor characteristics). pN (Lymph nodes characteristics). PL (Pleural invasion). STAS (Spread Through Air Spaces). BD: Tumor Budding group. Number in bold indicate statistically significant results.

**Table 4 cancers-14-02281-t004:** Multivariate analysis of survival for clinical and pathological parameters.

Variable	IC95%	HR	*p* (Pr > z)
**PS**			
1–2	0.37–1.38	0.72	0.324
3–4	0.81–5.06	2.02	0.131
**Location (peripheral or intermediate)**	0.39–1.47	0.75	0.41
**pT**			
pT2	0.60–3.17	1.38	0.452
pT3	0.30–3.34	1.01	0.991
pT4	0.49–13.18	2.55	0.263
**pN**			
pN1	0.19–3.62	0.83	0.807
pN2	0.66–21.78	3.80	0.133
**Pathological Stage**			
II	0.80–5.46	2.09	0.134
III	0.12–4.62	0.75	0.758
**Number of metastatic lymph nodes station**			
1	0.29–3.07	0.95	0.934
≥2	NA	NA	NA
**Pleural invasion**			
PL1-PL2	0.73–3.1	1.51	0.263
PL3	0.18–3.62	0.80	0.778
**Histopathological subtype**			
Not keratinizing	0.47–1.66	0.88	0.698
Basaloid	0.02–1.38	0.018	0.098
**Percentage of tumor necrosis**			
≥30% et <50%	0.57–3.21	1.35	0.495
≥50%	0.79–3.41	1.64	0.186
**Vascular and/or lymphatic emboli present**	0.64–2.89	1.36	0.418
**Single cell invasion present**	0.56–7.37	2.03	0.358
**STAS**			
≤3 alveoli	0.23–1.36	0.55	0.2
>3 alveoli	1.18–6.33	2.74	**0.018**
**BD**			
BD1	0.71–4.96	1.87	0.207
BD2	0.31–3.04	0.97	0.968
BD3	0.37–4.3	1.26	0.71
**Cell nests group ≥ 5**			
≥1–<5	0.44–4.50	1.41	0.565
1	NA	NA	NA

PS: Performance Status. pT (Primary tumor characteristics). pN (Lymph nodes characteristics). PL (Pleural invasion). STAS (Spread Through Air Spaces). BD: Tumor Budding group. NA: Not Available. Number in bold indicate statistically significant results.

**Table 5 cancers-14-02281-t005:** Clinical and histopathological data in relation to five-year overall survival.

Variable	n (%)	5-Year Survival Rate	*p* (Log-Rank Test)
**Age**			0.47
≤65 years	60 (36.8)	63.3%	
>65 years	103 (63.2)	70.9%	
**PS**			
0	56 (34.4)	46.4%	0.20
1–2	54 (33.1)	48.2%	
3–4	9 (5.5)	11.1%	
Unknown	44 (27)	45.4%	
**pT**			**0.006**
pT1	71 (45.6)	60.6%	
pT2	67 (41.1)	34.3	
pT3	25 (15.3)	28%	
**pN**			**0.72**
pN0	130 (79.7)	48.5%	
pN1	33 (20.2)	30.3%	
**Pathological Stage**			**0.019**
I	91 (55.8)	56%	
II	70 (42.9)	31.4%	
**Pleural invasion**			**0.23**
PL0	127 (77.9)	46.4%	
PL1 or PL2	29 (17.8)	41.4%	
PL 3	7 (4.3)	28.6%	
**Percentage of tumor necrosis**			**0.009**
<30%	104 (63.8)	61.5%	
≥30% et <50%	27 (16.6)	11.%	
≥50%	32 (19.6)	18.7%	
**Vascular and/or lymphatic emboli**			**0.15**
Absent	135 (82.8)	49.2%	
Present	28 (17.2)	28.6%	
**STAS**			0.22
Absent	104 (63.8)	42.3%	
≤3 alveoli	30 (18.4)	50%	
>3 alveoli	28 (17.2)	50%	
Not assessable			
**STAS**			0.97
Absent	104 (63.8)	42.3%	
≤1 mm	37 (22.7)	45.9%	
>1 mm	22 (13.5)	54.5%	
**Budding**			**0.061**
Zero	56 (34.4)	57.1%	
BD1	57 (35)	36.8%	
BD2	31 (19)	38.7%	
BD3	17 (10.4)	35.3%	
Not assessable	6 (3.7)	33.3%	
**Single cell invasion**			**0.11**
Absent	112 (68.7)	52.7%	
Present	51 (31.3)	27.4%	
**Cell nests group**			**0.062**
1	51 (31.3)	27.4%	
≥1–<5	72 (44.2)	45.8%	
≥5	39 (23.9)	62.3%	
Not assessable	1 (0.6)	100%	
**Mitoses**			0.53
<20/2 mm^2^	138 (64.7)	43.5%	
≥20/2 mm^2^	24 (14.7)	50%	
Not assessable	1 (0.6)	100%	

PS: Performance Status. pT (Primary tumor characteristics). pN (Lymph nodes characteristics). PL (Pleural invasion). STAS (Spread Through Air Spaces). BD: Tumor Budding group. Number in bold indicate statistically significant results.

## Data Availability

The datasets generated and analyzed during the current study are not publicly available due the rules of the European General Data Protection Regulation but are available from the corresponding author on reasonable request.

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
