# Peer review of "Spread Through Air Spaces (STAS) Is an Independent Prognostic Factor in Resected Lung Squamous Cell Carcinoma"

_cancers, 2022, doi:10.3390/cancers14092281_

Round 1
Reviewer 1 Report
Recently I was invited to review an interesting paper entitled: Spread Through Air Spaces (STAS) is an independent prognostic factor in resected lung squamous cell carcinoma. It is important to discuss novel prognostic factors in an early NSCLC. Especially, squamous cell carcinoma is an interesting target of this kind of retrospective studies. The paper is in general well written. However, there are some typos that require editing. Still, I have some remarks that require addressing by the authors.
Major remarks
Line 157. The authors include the patients who underwent the wedge excision and segmentectomies in the paper analyzing the influence of the presence of STAS on the prognosis in patients with an early NSCLC. I believe that this extent of the resection may influence the possibility of the assessment of the presence of STAS and the distance of the STAS site from the primary tumor. Moreover, this extent of the resection may be clearly insufficient in the presence of STAS. Please consider removing this small group from the analyzed cohort.
Table 1. Please define the term “in situ carcinoma”. Did it apply to the diameter/infiltration of the tumor? This term is misleading- did the authors mean Tis?
Table 1. Please define the term "obstructive pneumopathy". Did the authors mean COPD?
Table 1. Please define the term "peritumoral pneumopathy".
Table 1. What does it mean side D vs. G?
Figure 1. This figure is too busy. Please consider reducing the number of graphs included. The graph “d” should be rather: “Stage” instead of “Tumor stage”. This kind of busy figure decreases the technical possibility of adding the tables with numbers at risk. Those tables are an immanent part of survival analysis and should be included. Moreover, it would be more clear to present the graphs of the features which gained significance in the multivariate analysis.
Lines 157-158. How many patients were lost to follow-up? What were the methods of follow-up?
Tables 3-4. The evaluated factor: “Presence of STAS” did not prove to be a prognostic factor in the univariate analysis. However, it was proved to be an independent prognosticator in the multivariate analysis. I am not a statistician but I would like to experience a wider comment on this part of the statistical analysis as the factors which do not reach the level of statistical significance (eventually have a p-value over 0.1 or 0.2) are excluded from the analysis.
Lines 310-316. In my opinion, a significant limitation of the study is the fact that the endpoint was OS. In this kind of analysis, the endpoint with maximal statistical significance is DFS. It is obvious that DFS is more difficult to obtain. Please comment on that.
It would be interesting to see the subgroup analysis – for example limited to stage I-II, in patients who underwent lobectomy. Including very strong prognosticators like PS 3-4, pN2, or pneumonectomy in an analysis of a subtle issue of STAS makes the analysis and further conclusion more difficult.
Minor remarks
Table 3. There are a few typos in the table. There is also an inconsistent number of decimal places in the p-value section. Please correct.
Please explain the abbreviations below tables and figures.
Author Response
We would like to thank the reviewers for their insightful comments. We have chosen to address the points raised and resubmit this manuscript.
Recently I was invited to review an interesting paper entitled: Spread Through Air Spaces (STAS) is an independent prognostic factor in resected lung squamous cell carcinoma. It is important to discuss novel prognostic factors in an early NSCLC. Especially, squamous cell carcinoma is an interesting target of this kind of retrospective studies. The paper is in general well written. However, there are some typos that require editing. Still, I have some remarks that require addressing by the authors.
Major remarks
Line 157. The authors include the patients who underwent the wedge excision and segmentectomies in the paper analyzing the influence of the presence of STAS on the prognosis in patients with an early NSCLC. I believe that this extent of the resection may influence the possibility of the assessment of the presence of STAS and the distance of the STAS site from the primary tumor. Moreover, this extent of the resection may be clearly insufficient in the presence of STAS. Please consider removing this small group from the analyzed cohort. -> STAS was present in 8/19 (42%) cases of patients with wedge resection/segmentectomy, whereas it was present in 74/220 (34%) cases of patients with lobectomy/pneumonectomy. Thus, the type of resection does not seem to influence the presence of STAS. Since the size of a wedge resection is several centimeters, it seems unlikely that the type of resection influences the presence or absence of STAS. Therefore, we did not delete these patients. Finally, patients with squamous cell carcinoma often have comorbidities and may not benefit from lobectomy/pneumonectomy. Including wedge resections or segmentectomies represents more of a real-life condition.
Table 1. Please define the term “in situ carcinoma”. Did it apply to the diameter/infiltration of the tumor? This term is misleading- did the authors mean Tis? -> We have modified and added the following sentences : “The presence of carcinoma in situ was recorded on the adjacent bronchus. Carcinoma in situ was not considered for the measurement of tumor diameter.”
Table 1. Please define the term "obstructive pneumopathy". Did the authors mean COPD? -> We have modified the term “obstructive pneumopathy” to obstructive pneumonitis whichis the term used in TNM staging. We have added the following sentence in Material and Methods section.
Table 1. Please define the term "peritumoral pneumopathy". -> We have added the following sentence : “We recorded the presence of peritumoral pneumopathy defined by the presence of pneumopathy lesions in contact with the tumor without an obstructive lesion in a bronchus.”
Table 1. What does it mean side D vs. G? -> We apologize for this mistake, we have modified D and G to Right and Left in Table 1.
Figure 1. This figure is too busy. Please consider reducing the number of graphs included. The graph “d” should be rather: “Stage” instead of “Tumor stage”. This kind of busy figure decreases the technical possibility of adding the tables with numbers at risk. Those tables are an immanent part of survival analysis and should be included. Moreover, it would be more clear to present the graphs of the features which gained significance in the multivariate analysis. -> We do not want to delete the graphs. Indeed, each of them brings important information, especially concerning the validity of our cohort. We believe that limiting Figure 1 only to factors significant in multivariate analysis would be too restrictive and the manuscript would lose some information that is important to the reader.
Lines 157-158. How many patients were lost to follow-up? What were the methods of follow-up? -> We have added the following sentence : “Twenty-one patients were lost to follow-up.” This is a retrospective study, the follow-up is the usual clinical follow-up of lung cancers.
Tables 3-4. The evaluated factor: “Presence of STAS” did not prove to be a prognostic factor in the univariate analysis. However, it was proved to be an independent prognosticator in the multivariate analysis. I am not a statistician but I would like to experience a wider comment on this part of the statistical analysis as the factors which do not reach the level of statistical significance (eventually have a p-value over 0.1 or 0.2) are excluded from the analysis.
Lines 310-316. In my opinion, a significant limitation of the study is the fact that the endpoint was OS. In this kind of analysis, the endpoint with maximal statistical significance is DFS. It is obvious that DFS is more difficult to obtain. Please comment on that. -> We have added the following sentences in the discussion section : “Another limitation of our work is that our primary endpoint is 5-year survival; we could have assessed progression-free survival. Nevertheless, overall survival is a more important criterion than progression-free survival in the evaluation of prognosis in lung cancer.”
It would be interesting to see the subgroup analysis – for example limited to stage I-II, in patients who underwent lobectomy. Including very strong prognosticators like PS 3-4, pN2, or pneumonectomy in an analysis of a subtle issue of STAS makes the analysis and further conclusion more difficult. -> We conducted a subgroup analysis suggested in Chapter 3.5. We have added Table 5 and discussed the limitations of these results in the discussion section.
Minor remarks
Table 3. There are a few typos in the table. There is also an inconsistent number of decimal places in the p-value section. Please correct. -> We apologize for this; we have corrected the decimal in p-value sections.
Please explain the abbreviations below tables and figures. -> We have explained the abbreviations below tables and figures.
Reviewer 2 Report
The authors (Dagher et al.) searched for morphological histopathological prognostic factors in lung squamous cell carcinoma (LUSC) by performing single-center retrospective study of 241 LUSC patients. They found that extensive Spread Through Air Spaces (STAS) is an independent factor of poor prognosis in LUSC. Moreover, they concluded that STAS is correlated with the presence of other poor prognostic factors such as emboli, pleural invasion and would reflect a greater tumor aggressiveness.
Histopathological factors were extensively investigated for prognostic impact on 5-year survival in this study, however, there are some weaknesses as shown below.
- STAS >3 alveoli was shown as an independent factor of poor prognosis based on multivariate analysis. However, the STAS >3 alveoli was not related to overall survival on univariate analysis (Table 3). Moreover, Kaplan Meier analysis for 5-year overall survival showed that survival rate was not significantly different among STAS status (Table 1 and Figure 1).
- The authors insisted that STAS >3 alveoli, a parameter of extensive STAS, was shown as an independent factor of poor prognosis, however, STAS >1 mm, another parameter of extensive STAS, was not shown as an independent factor of poor prognosis. Although the authors described the reason for the discrepancy in the Discussion section, I think it is difficult to believe the reason.
- Line 187, Fig 2 should be revised to Fig 1. Also Fig 2 should be correctly cited.
- How many slide glasses for LUSC per case were examined?
- In abstract, Spread Through Air Spaces (STAS) should be included.
- Tables 3 and 4: Titles are too short.
- There are some minor mistakes:
- Line 14: histopronostic -> histoprognostic
- Table 1: obstrutive pneumopathy -> obstructive pneumopathy
- Table 1 and Table 2: Differenciation -> Differentiation
- Line 281: signicativity -> significativity
Author Response
We would like to thank the reviewers for their insightful comments. We have chosen to address the points raised and resubmit this manuscript.
The authors (Dagher et al.) searched for morphological histopathological prognostic factors in lung squamous cell carcinoma (LUSC) by performing single-center retrospective study of 241 LUSC patients. They found that extensive Spread Through Air Spaces (STAS) is an independent factor of poor prognosis in LUSC. Moreover, they concluded that STAS is correlated with the presence of other poor prognostic factors such as emboli, pleural invasion and would reflect a greater tumor aggressiveness.
Histopathological factors were extensively investigated for prognostic impact on 5-year survival in this study, however, there are some weaknesses as shown below.
- STAS >3 alveoli was shown as an independent factor of poor prognosis based on multivariate analysis. However, the STAS >3 alveoli was not related to overall survival on univariate analysis (Table 3). Moreover, Kaplan Meier analysis for 5-year overall survival showed that survival rate was not significantly different among STAS status (Table 1 and Figure 1).
- The authors insisted that STAS >3 alveoli, a parameter of extensive STAS, was shown as an independent factor of poor prognosis, however, STAS >1 mm, another parameter of extensive STAS, was not shown as an independent factor of poor prognosis. Although the authors described the reason for the discrepancy in the Discussion section, I think it is difficult to believe the reason. -> We have discussed the limits of our study and expanded the limits of our work in the discussion section.
- Line 187, Fig 2 should be revised to Fig 1. Also Fig 2 should be correctly cited. -> We apologize for the mistake, we have modified Fig.2 to Fig.1
- How many slide glasses for LUSC per case were examined? -> We have added the following sentence : “The mean number of slides examined per case was 6.2 +/-0.1.”
- In abstract, Spread Through Air Spaces (STAS) should be included.-> We have added this line 19-20.
- Tables 3 and 4: Titles are too short. -> We have expanded the titles of Tables 3 & 4
- There are some minor mistakes:
- Line 14: histopronostic -> histoprognostic
- Table 1: obstrutive pneumopathy -> obstructive pneumopathy
- Table 1 and Table 2: Differenciation -> Differentiation
- Line 281: signicativity -> significativity
- We apologize for these mistakes, we have corrected them.
Round 2
Reviewer 1 Report
I would like to thank the authors for the efforts they have made to improve their text. They have commented on nearly all of my remarks. Despite some of the remarks did not result in any correction in the text, I still think it is an acceptable approach toward the review in a scientific paper. However, I have commented on one issue that was not addressed in any way:
"Tables 3-4. The evaluated factor: “Presence of STAS” did not prove to be a prognostic factor in the univariate analysis. However, it was proved to be an independent prognosticator in the multivariate analysis. I am not a statistician but I would like to experience a wider comment on this part of the statistical analysis as the factors which do not reach the level of statistical significance (eventually have a p-value over 0.1 or 0.2) are excluded from the analysis. "
Again, please comment on the methods, results, and discussion. The factor that did not gain statistical significance in the univariate analysis is a significant prognosticator in the multivariate. What kind of model was built to disclose this result. Please have in mind that the title is very bold: "Spread Through Air Spaces (STAS) is an independent prognostic factor in resected lung squamous cell carcinoma" - it must be supported by well-documented data.
"the follow-up is the usual clinical follow-up of lung cancers" - readers interested in the prognosis of NSCLC know what is usual clinical follow-up performed in one oncological center is. My intention was to be informed if the national databases were used to define the follow-up.
Minor remarks. There are still typos in the tables.
Author Response
We would like to thank the #1 reviewer for his/her comments. We have chosen to respond to the comments and modify the manuscript according to the comments. We hope this has improved the quality of the manuscript.
I would like to thank the authors for the efforts they have made to improve their text. They have commented on nearly all of my remarks. Despite some of the remarks did not result in any correction in the text, I still think it is an acceptable approach toward the review in a scientific paper. However, I have commented on one issue that was not addressed in any way:
"Tables 3-4. The evaluated factor: “Presence of STAS” did not prove to be a prognostic factor in the univariate analysis. However, it was proved to be an independent prognosticator in the multivariate analysis. I am not a statistician but I would like to experience a wider comment on this part of the statistical analysis as the factors which do not reach the level of statistical significance (eventually have a p-value over 0.1 or 0.2) are excluded from the analysis. "
Again, please comment on the methods, results, and discussion. The factor that did not gain statistical significance in the univariate analysis is a significant prognosticator in the multivariate. What kind of model was built to disclose this result. Please have in mind that the title is very bold: "Spread Through Air Spaces (STAS) is an independent prognostic factor in resected lung squamous cell carcinoma" - it must be supported by well-documented data. -> We have modified the following sentence in the methods section : « Prognostic significance of clinical and pathological characteristics with p<0.25 with log rank test were analyzed with univariate Cox regression and multivariate Cox model to determine hazard ratios (HRs).”
We have added the following sentence in the results section : “In univariate and multivariate analysis, only parameters with a p<0.25 with the log-rank test were retained for the univariate test. The multivariate analysis retained all parameters with a p<0.25 with the log-rank test in order to retain all parameters with a potential role, even below the significance thresholds, in order to best correlate the different parameters including STAS to prognosis.”
We have added the following sentence in the discussion section : “Our statistical analysis, shows no correlation in univariate analysis, but shows a correlation between survival and the presence of STAS in multivariate analysis. As multivariate analysis allows to take into account the weight of different parameters, it allows to correct the weight given to some histological parameters. Moreover, our approach allows to take into account parameters for which univariate analysis could lack statistical power because of a too small number of patients."
the follow-up is the usual clinical follow-up of lung cancers" - readers interested in the prognosis of NSCLC know what is usual clinical follow-up performed in one oncological center is. My intention was to be informed if the national databases were used to define the follow-up. -> We did not use the national databases, we have added the following sentence in the discussion section : « In our work, the 5-year overall survival is comparable to other studies: 70% for stage I, 57% in stage II and 21% in stage III/IV in the work of Kadota et al [17]. »
Minor remarks. There are still typos in the tables. -> We apologize for the inconvenience, we have corrected typos in the tables.
Reviewer 2 Report
I think that the revision by the authors is insufficient as shown below. Thus, I recommend “Rejection”.
- “STAS >3 alveoli was shown as an independent factor of poor prognosis based on multivariate analysis. However, the STAS >3 alveoli was not related to overall survival on univariate analysis (Table 3). Moreover, Kaplan Meier analysis for 5-year overall survival showed that survival rate was not significantly different among STAS status (Table 1 and Figure 1).”
To the above comment, there was no reply.
- “Line 187, Fig 2 should be revised to Fig 1. Also Fig 2 should be correctly cited.”
To the above comment, the revision is insufficient. Figure 2 is not cited in the text.
Author Response
We would like to thank the #2 reviewer for his/her comments. We have chosen to respond to the comments and modify the manuscript according to the comments. We hope this has improved the quality of the manuscript.
I think that the revision by the authors is insufficient as shown below. Thus, I recommend “Rejection”.
“STAS >3 alveoli was shown as an independent factor of poor prognosis based on multivariate analysis. However, the STAS >3 alveoli was not related to overall survival on univariate analysis (Table 3). Moreover, Kaplan Meier analysis for 5-year overall survival showed that survival rate was not significantly different among STAS status (Table 1 and Figure 1).” -> -> We discussed this point as also stated by reviewer #1. We have modified the following sentence in the methods section : « Prognostic significance of clinical and pathological characteristics with p<0.25 with log rank test were analyzed with univariate Cox regression and multivariate Cox model to determine hazard ratios (HRs).”
We have added the following sentence in the results section : “In univariate and multivariate analysis, only parameters with a p<0.25 with the log-rank test were retained for the univariate test. The multivariate analysis retained all parameters with a p<0.25 with the log-rank test in order to retain all parameters with a potential role, even below the significance thresholds, in order to best correlate the different parameters including STAS to prognosis.”
We have added the following sentence in the discussion section : “Our statistical analysis, shows no correlation in univariate analysis, but shows a correlation between survival and the presence of STAS in multivariate analysis. As multivariate analysis allows to take into account the weight of different parameters, it allows to correct the weight given to some histological parameters. Moreover, our approach allows to take into account parameters for which univariate analysis could lack statistical power because of a too small number of patients."
To the above comment, there was no reply.
- “Line 187, Fig 2 should be revised to Fig 1. Also Fig 2 should be correctly cited.” -> We have cited Figure 2 in the text and modified the position of Fig.1 and Fig.2.
To the above comment, the revision is insufficient. Figure 2 is not cited in the text.
Round 3
Reviewer 2 Report
The authors added the following sentences in the method section; “In univariate and multivariate analysis, only parameters with a p<0.25 with the log-rank test were retained for the univariate test. The multivariate analysis retained all parameters with a p<0.25 with the log-rank test in order to retain all parameters with a potential role, even below the significance thresholds, in order to best correlate the different parameters including STAS to prognosis.” Is this popular method? I do not think so. I would like to recommend “Rejection”.
In line 204, “The correlation between clinical data and survival is shown in Table 1 and Fig.122.” is described. What does this mean?